# AI-Enhanced Problem-Based Learning for Sustainable Engineering Education: The AIPLE Framework for Developing Countries

Romain Kazadi Tshikolu [1], David Kule Mukuhi [1], Tychique Nzalalemba Kabwangala [1], Jonathan Ntiaka Muzakwene [1] and Anderson Sunda-Meya [2,*]

1   Faculté des Sciences et Technologies, Université Loyola du Congo, Kinshasa B.P. 7245,
    Democratic Republic of the Congo; ktshikolusj@gmail.com (R.K.T.); mr.davidkule@gmail.com (D.K.M.);
    tychnzal15@gmail.com (T.N.K.); jonathantiaka@gmail.com (J.N.M.)
2   Department of Physics, Xavier University of Louisiana, New Orleans, LA 70125, USA
*   Correspondence: asundame@xula.edu

## Abstract

Engineering education in developing countries faces critical challenges that hinder progress toward achieving the United Nations Sustainable Development Goals (SDGs). In the Democratic Republic of Congo (DRC), students entering engineering programs often exhibit significant apprehension toward foundational sciences, creating barriers to developing the technical competencies required for sustainable development. This paper introduces the AI-Integrated Practical Learning in Engineering (AIPLE) Framework, an innovative pedagogical model that synergizes Problem-Based Learning (PBL), hands-on experimentation, and strategic Artificial Intelligence (AI) integration to transform engineering education for sustainability. The AIPLE framework employs a five-stage cyclical process designed to address student apprehension while fostering sustainable engineering mindsets essential for achieving SDGs 4 (Quality Education), 7 (Affordable and Clean Energy), 9 (Industry, Innovation and Infrastructure), and 11 (Sustainable Cities and Communities). This study, grounded in qualitative surveys of engineering instructors at Université Loyola du Congo (ULC), demonstrates how the framework addresses pedagogical limitations while building technical competency and sustainability consciousness. The research reveals that traditional didactic methods inadequately prepare students for complex sustainability challenges, while the AIPLE framework's integration of AI-assisted learning, practical problem-solving, and sustainability-focused projects offers a scalable solution for engineering education transformation in resource-constrained environments. Our findings indicate strong instructor support for PBL methodologies and cautious optimism regarding AI integration, with emphasis on addressing infrastructure and ethical considerations. The AIPLE framework contributes to sustainable development by preparing engineers who are technically competent and committed to creating environmentally responsible, socially inclusive, and economically viable solutions for developing countries.

**Keywords:** engineering education; sustainable development; artificial intelligence; problem-based learning; developing countries; curriculum innovation; pedagogical innovation; Sub-Saharan Africa; sustainability education; SDGs

# 1. Introduction

## 1.1. Engineering Education and Sustainable Development: A Critical Nexus

The achievement of the United Nations Sustainable Development Goals (SDGs) by 2030 requires a fundamental transformation in how engineers are educated, particularly in developing countries where the challenges are most acute and the engineering workforce gaps most pronounced [1]. Engineering education serves as a critical catalyst for sustainable development, with engineers playing essential roles in addressing climate change (SDG 13), ensuring access to clean energy (SDG 7), building sustainable infrastructure (SDG 9), and creating sustainable cities and communities (SDG 11) [2]. However, traditional engineering education models, particularly in the Sub-Saharan Africa, often fail to adequately prepare students for the complex, interdisciplinary challenges of sustainable development [3].

In sub-Saharan Africa, an estimated 2.5 million additional engineers are needed to address the region's pressing development challenges, from constructing climate-resilient infrastructure to developing sustainable energy systems [4]. The Democratic Republic of Congo (DRC), with its vast natural resources and urgent infrastructure needs, exemplifies both the potential and the challenges facing engineering education in developing countries. The nation's engineering workforce requirements are critical for achieving sustainable development, yet educational institutions struggle with resource constraints, outdated pedagogical approaches, and student preparation challenges that limit their effectiveness [5].

## 1.2. The Student Apprehension Challenge: Barriers to Sustainable Engineering Education

Despite the critical demand for engineers, engineering education in the DRC confronts a fundamental paradox. While students are motivated to pursue engineering careers, their pre-university preparation creates a deep schism within the classroom, fostering a culture of passive learning that stifles the development of essential engineering competencies. The former typically possess strong theoretical backgrounds in mathematics and physics but lack practical, hands-on skills, while the latter exhibits the opposite imbalance, being more adept at technical application but weaker in foundational theory [6].

This heterogeneity creates a challenging and often disengaging learning environment. Students with strong theoretical knowledge become "annoyed" by the remedial pace required to support their peers, while those lacking foundational concepts show "early signs of demotivation and discouragement". This foundational insecurity from both ends of the spectrum manifests in passive learning behaviors that are antithetical to the active, critical thinking required for sustainable engineering practice. Instructors observe that students "wait for 'recipes' to apply in solving problems" and tend to "memorize solutions of class examples" rather than developing the adaptive problem-solving skills essential for addressing novel challenges. This "recipe-seeking" mindset is a rational response to a fractured classroom environment, as it represents the lowest common denominator for passing examinations without requiring genuine conceptual understanding [7]. Such a passive approach is particularly problematic for sustainable engineering, which demands systems thinking, interdisciplinary collaboration, and innovative approaches to complex, context-specific problems [8].

## 1.3. The AIPLE Framework: A Sustainable Engineering Education Solution

This paper introduces the AI-Integrated Practical Learning in Engineering (AIPLE) Framework, a novel pedagogical model specifically designed to address the confidence and mindset gaps that hinder effective engineering education for sustainable development. The AIPLE framework represents a paradigm shift from traditional didactic instruction to an active, student-centered approach that integrates three key components:

1. Problem-Based Learning (PBL) grounded in real-world sustainability challenges.

2. Hands-on experimentation that connects theory to practical application.
3. Strategic AI integration that provides personalized learning support and enhances problem-solving capabilities.

The framework is explicitly designed to advance sustainable engineering education by developing the technical competencies required for SDG achievement, fostering sustainability mindsets, building student confidence in tackling complex problems, and providing scalable solutions for resource-constrained educational environments.

### 1.4. Research Objectives and Contributions

This study aims to:

1. Examine the limitations of traditional engineering education in developing sustainable engineering competencies.
2. Present the AIPLE framework as a comprehensive solution for sustainable engineering education.
3. Analyze instructor perspectives on PBL and AI integration in engineering education.
4. Demonstrate the framework's alignment with sustainable development objectives.
5. Provide implementation guidelines for similar institutions in developing countries.

The research contributes to the field by offering a theoretically grounded, practically oriented framework that addresses both pedagogical and sustainability challenges in engineering education, with particular relevance for developing countries working toward SDG achievement.

## 2. Literature Review

### 2.1. Engineering Education for Sustainable Development

Engineering Education for Sustainable Development (EESD) has emerged as a critical field addressing the need to prepare engineers who can contribute to achieving the SDGs [9]. The UNESCO Engineering for Sustainable Development report emphasizes that engineering is crucial for achieving nearly all 17 SDGs, requiring educational approaches that integrate technical competency with sustainability consciousness [1]. However, traditional engineering education often fails to develop the interdisciplinary thinking, systems perspective, and ethical reasoning required for sustainable engineering practice [10].

Research highlights the importance of active learning methodologies in developing sustainability competencies. Mathebula (2018) argues for a capabilities approach to engineering education that emphasizes human development and social responsibility [10]. Recent scholarship underscores the urgency of adopting transformative, problem-oriented pedagogies to cultivate these competencies. A 2024 literature review confirms that the use of PBL as a methodology for sustainability education is a relatively recent innovation, with a strong preference for its integration into engineering courses [11]. Such approaches are seen as essential for developing key sustainability competencies, including systems thinking, interpersonal skills, and integrated problem-solving. Despite this potential, the integration of PBL for sustainability in higher education often lacks a systemic perspective and remains under-explored, particularly in the context of developing countries. The literature shows that sustainability is frequently treated as an "add-on" rather than being deeply embedded within core engineering problems, which limits its effectiveness [12]. The AIPLE framework addresses this gap by positioning authentic sustainability challenges as the central driver of the learning process.

### 2.2. Problem-Based Learning in Engineering Education

Problem-Based Learning (PBL) has gained recognition as an effective pedagogical approach for engineering education, particularly in developing critical thinking and problem-

solving skills [13]. PBL's emphasis on real-world problems, collaborative learning, and student-centered instruction aligns well with the competencies required for sustainable engineering practice [14]. Research demonstrates that PBL can improve student engagement, retention, and preparation for professional practice [15].

However, PBL implementation in developing countries faces unique challenges, including resource constraints, large class sizes, and instructor preparation [16]. Studies from similar contexts in Africa highlight the importance of adapting PBL methodologies to local conditions while maintaining pedagogical effectiveness [17]. The AIPLE framework addresses these challenges by providing structured implementation guidelines and leveraging technology to support PBL delivery in resource-constrained environments.

### 2.3. Artificial Intelligence in Engineering Education

The integration of AI in education has shown promising results for personalizing learning, providing intelligent tutoring, and enhancing student engagement [18]. In engineering education specifically, AI applications include adaptive learning systems, automated assessment tools, and simulation environments that support hands-on learning [19]. Recent studies demonstrate AI's potential to address individual learning needs and provide scaffolding for complex problem-solving [20].

However, AI implementation in developing countries faces significant challenges, including infrastructure limitations, digital divides, and ethical considerations [21]. Research emphasizes the importance of contextually appropriate AI integration that addresses local needs while avoiding technological dependency [22–25]. This complex reality necessitates a cautious and context-aware approach. The AIPLE framework is therefore positioned not as a techno-utopian solution but as a pragmatic model that acknowledges these barriers. Its emphasis on "strategic" and "appropriate" AI integration is a direct response to the need for sustainable and equitable technological adoption in the sub-Saharan Africa.

## 3. Methodology

### 3.1. Research Design

This study employs a qualitative research approach to explore instructor perspectives on engineering education challenges and the potential of the AIPLE framework. The research was conducted at Université Loyola du Congo (ULC) in Kinshasa, Democratic Republic of Congo, between January and July 2025. The study received ethical approval from the ULC Research Ethics Committee and followed established guidelines for educational research.

### 3.2. Participants

This study was conducted at Université Loyola du Congo (ULC), a nascent university where the engineering program is in its early stages of development. The study's participants comprised the four permanent, full-time instructors of the Faculté des Sciences et Technologies. As the core faculty, these four instructors are responsible for delivering the entirety of the foundational engineering curriculum, giving them a unique and comprehensive vantage point on the pedagogical challenges and student needs within this specific educational environment.

Therefore, the selection of participants, while purposive, also represents a complete census of the permanent engineering faculty at the institution. The instructors have extensive experience in engineering education (ranging from 5 to 15 years) and have demonstrated active engagement with pedagogical innovation. Their collective expertise spans diverse disciplines, including mechanical, electrical, and computer science, ensuring a broad and well-rounded perspective on teaching foundational engineering courses.

*3.3. Data Collection*

Data were collected through a semi-structured written survey administered to each participant. The survey instrument was designed to elicit detailed responses across three main areas: (1) current teaching practices and challenges, (2) experiences with and perspectives on Problem-Based Learning, and (3) views on the potential and risks of AI integration in engineering education. The survey included a mix of open-ended and targeted questions to capture rich, detailed qualitative data from the instructors' professional experiences. The full survey instrument is provided in Appendix A.

*3.4. Data Analysis*

Qualitative data were analyzed using thematic analysis, following Braun and Clarke's (2006) six-phase approach [26]. Key themes were identified through iterative coding and constant comparison. Quantitative data from closed-ended questions were analyzed using descriptive statistics. The analysis focused on identifying patterns in instructor experiences and perspectives that inform the AIPLE framework design.

*3.5. Limitations*

This study's limitations include its small sample size (four instructors) and its focus on a single institution. As such, the findings are not intended to be generalizable but rather to provide a deep, context-rich exploration of the issues at hand. The study also focuses exclusively on instructor perspectives; future research should incorporate student viewpoints to provide a more holistic understanding of the learning experience. Additionally, this paper presents a theoretical framework without empirical validation of student outcomes, which represents a critical and necessary next step for future research.

## 4. The AIPLE Framework: A Context-Adapted Pedagogical Model

*4.1. Theoretical Foundation*

The AIPLE framework is grounded in constructivist learning theory, which posits that learners actively construct knowledge through experience and reflection rather than passively receiving information. It integrates three complementary theoretical approaches to create a holistic learning environment:

1.  Experiential Learning Theory: The framework's five-stage cyclical process mirrors Kolb's cycle of concrete experience, reflective observation, abstract conceptualization, and active experimentation, ensuring that learning is an iterative and deeply embedded process [27].
2.  Social Constructivist Theory: AIPLE emphasizes learning through social interaction and collaborative problem-solving. Students work in diverse teams, engaging in peer-to-peer learning and co-constructing knowledge, which is particularly effective for bridging the skill gaps present in the classroom [28].
3.  Education for Sustainable Development (EESD) Theory: The framework embeds sustainability principles into the core learning process, moving beyond treating sustainability as an add-on topic. It encourages students to develop the systems thinking and ethical reasoning necessary to address complex, real-world sustainability challenges [29].

*4.2. The Five Stages of AIPLE: An Implementation and Assessment Guide*

The AIPLE framework consists of five interconnected stages that form a cyclical learning process. It is designed to be a practical tool for educators, providing a clear structure for implementation and assessment. The iterative nature of the framework, particularly the loop between Stages 2, 3, and 4, allows for continuous refinement and deeper learning as

students cycle through research, conceptualization, and experimentation until the project goals are met. Table 1 provides a detailed guide to implementing the framework.

**Table 1.** AIPLE Framework: Activities, AI Integration, and Assessment Methods.

| Stage | Objective | Key Activities & Instructor Examples | Strategic AI Integration | Assessment Methods (Formative & Summative) |
|---|---|---|---|---|
| Stage 1: Problem Identification & Contextualization | Present real-world sustainability challenges relevant to the local context, connecting them to foundational scientific principles. | - **Identify community-based problems** related to SDGs (e.g., clean energy, sustainable housing) through engagement with local stakeholders.<br>- **Frame the problem:** The instructor presents a real-world challenge, such as designing a semi-automatic system for making fired briquettes.<br>- **Brainstorming & Division:** Students brainstorm and divide the macro-problem into manageable sub-problems for different teams (e.g., raw material research, mechanical design, electrical systems). | - **AI-Powered Problem Databases:** Use AI tools to search for and curate case studies and real-world engineering problems relevant to local SDG challenges.<br>- **Context Analysis Tools:** Employ AI to analyze local data (e.g., environmental reports, community surveys) to help contextualize the problem. | - **Formative:** Group discussion participation; initial problem statement draft.<br>- **Summative:** Finalized project proposal outlining the problem, scope, and objectives. |
| Stage 2: Collaborative Investigation & Research | Develop research, analytical, and teamwork skills through guided and independent inquiry. | - **Individual & Group Research:** Students research existing systems, identify knowledge gaps, and explore theoretical concepts needed to solve their sub-problem.<br>- **Resource Consultation**: Students consult academic papers, technical manuals, and online resources. They can also consult with experts, such as local craftsmen or industry professionals.<br>- **Deliverables**: Teams submit study documents or research summaries to the instructor for feedback. | - **Intelligent Research Assistants:** Use AI tools like Elicit or Copilot to find, summarize, and synthesize relevant academic literature and technical documents.<br>- **Literature Analysis Tools:** Employ AI to identify key themes, methods, and gaps in a body of research. | - **Formative:** Research journal entries; annotated bibliography; peer review of research summaries.<br>- **Summative:** Comprehensive literature review and problem analysis report. |
| Stage 3: Theoretical Foundation & Concept Mapping | Connect the practical problem to underlying scientific and engineering principles, bridging the theory-practice gap. | - **Expert Input/Mini-Lectures:** The instructor provides targeted lectures or "expert time" on complex concepts that students struggle with, directly addressing knowledge gaps identified during research.<br>- **Concept Mapping**: Students create visual maps linking abstract theories (e.g., mechanics, electronics) to their practical application in the project.<br>- **Guided Discovery**: The instructor acts as a facilitator, guiding students to discover theoretical principles for themselves rather than providing "recipes". | - **Adaptive Tutoring Systems:** Use AI platforms that provide personalized exercises and explanations on foundational math and physics concepts based on student performance data.<br>- **Concept Visualization Tools:** Employ AI to generate diagrams, simulations, or interactive models that help students visualize complex engineering principles. | - **Formative:** Quizzes on foundational concepts; concept map submissions; participation in problem-solving sessions.<br>- **Summative:** Mid-project exam or presentation demonstrating mastery of relevant theoretical principles. |
| Stage 4: Hands-on Experimentation & Prototyping | Apply theoretical knowledge through practical experimentation, simulation, and prototype development. | -**Simulation:** Students use Computer Aided-Design (CAD)/Computer Aided-Engineering(CAE) software (e.g., LTSpice 24.0.12, MATLAB/SIMULINK 24.2) to design, simulate, and verify their solutions before building physical prototypes.<br>- **Workshop Sessions:** Students participate in hands-on workshops to learn practical skills (e.g., soldering, programming an Arduino) relevant to their project.<br>- **Prototyping:** Teams build real or virtual prototypes of their sub-systems (e.g., a manual brick-making machine, an electronic display).<br>- **Integration**: Teams meet regularly to integrate their sub-systems into a coherent final product. | - **Simulation Tools:** Leverage AI-enhanced simulation software (e.g., Ansys SimAI 2025 R2) to test complex systems and predict performance under various conditions.<br>- **AI-Assisted Coding:** Use tools like GitHub Copilot (v1.104) to assist with programming tasks, with a focus on understanding, modifying, and debugging the generated code.<br>- **Data Analysis Assistance:** Use AI to analyze experimental data, identify patterns, and visualize results. | - **Formative:** Lab notebook submissions; prototype demonstrations; code reviews.<br>- **Summative:** A functional real or virtual prototype; a technical report detailing the design, testing, and validation process. |

**Table 1.** *Cont.*

| Stage | Objective | Key Activities & Instructor Examples | Strategic AI Integration | Assessment Methods (Formative & Summative) |
|---|---|---|---|---|
| Stage 5: Reflection, Assessment & Iteration | Consolidate learning through reflection, peer evaluation, and presentation of the final solution to a wider audience. | - **Feedback & Iteration**: The instructor and peers provide feedback, and teams may be asked to make corrections or improvements.<br>- **Self & Peer Assessment**: Students reflect on their own learning process and the effectiveness of their team's collaboration.<br>- **Sustainability Evaluation:** Solutions are evaluated against sustainability criteria (environmental impact, social benefit, economic viability). | - **Automated Feedback Systems:** Use AI tools to provide initial feedback on written reports (e.g., for clarity, structure, plagiarism) or code (e.g., for efficiency, style).<br>- **Learning Analytics:** Instructors use AI dashboards to track student progress and engagement throughout the project, identifying individuals or teams that may need additional support. | - **Formative:** Reflective essays or journal entries; peer evaluation of teamwork.<br>- **Summative:** Final project presentation and demonstration; portfolio of work; final project report assessing the solution's sustainability impact. |

# 5. Results: Diverse Instructor Perspectives on Pedagogy and Innovation

The thematic analysis of the four instructor surveys revealed a rich and complex picture of the challenges, strategies, and evolving perspectives within engineering education at ULC. Three primary themes emerged: the foundational challenge of student apprehension and passive learning, the diverse and adaptive strategies for implementing PBL, and the nuanced views on AI integration, marked by both cautious optimism and significant practical hurdles.

## 5.1. The Core Challenge: Student Apprehension and the Culture of Passive Learning

All four instructors identified a deep-seated culture of passive learning as a primary obstacle. This culture is rooted in the previously described dichotomy of student preparation. Instructor KT noted that students often "wait for 'recipes' to apply in solving problems" and "memorize solutions of class examples," a behavior that actively hinders the development of adaptive problem-solving skills. Instructor JNM echoed this, describing a "student passivity" where many "expect to receive all knowledge from the teacher and rarely ask questions to challenge or engage". Instructor TNK added that a major challenge is that "students often do not prepare in advance," arriving in class without having engaged with the material beforehand.

This passive mindset is directly linked to the varied pre-university backgrounds. Instructors KT and JNM both described the challenge of a classroom containing students from scientific high schools, who are strong in theory but weak in practice, alongside students from technical high schools, who have the opposite skillset. This disparity creates a difficult teaching environment where it is challenging to "get the attention of all the students". The AIPLE framework's problem-first, hands-on approach is a direct pedagogical response to this challenge, designed to engage both groups by forcing the application of theory and the development of practical skills simultaneously.

## 5.2. Strategies in Practice: Adapting Problem- and Project-Based Learning

Despite the challenges, all instructors have independently moved toward active, project-based pedagogies, demonstrating a shared recognition of the limitations of traditional methods. However, their implementations vary significantly, reflecting adaptation to different course levels and learning objectives.

- Instructor KT employs a year-long, multi-group PBL project for first-year students in "Introduction to Industrial Engineering Science." A complex, real-world problem, such

as designing a briquette-making machine, is broken down into seven sub-problems (raw materials, mechanical design, electrical circuits, etc.), with each group tackling one piece. The process is iterative, involving a loop of research, expert consultation, and integration until a functional prototype is presented. This model is designed to build foundational skills and teamwork from the very beginning of the engineering curriculum.

- Instructor JNM utilizes a more formally structured, eight-phase PBL model for projects like designing a traffic light system. The phases—Go-Phase, Individual Research, Guidance, Collaborative Research, Presentation, Knowledge Sharing, Evaluation, and Feedback—provide a clear and rigorous process for students to follow. This highly structured approach ensures that all key aspects of problem-solving, from initial definition to final reflection, are systematically addressed.
- Instructor TNK, working with Master's students, implements Project-Based Learning focused on research. Students are assigned topics related to real-world challenges, such as energy issues in the DRC, and are guided through a phased research project with set milestones for progress tracking and feedback. This approach develops advanced research skills, often leading to an initial draft of a scientific paper, and teaches students to use professional tools like LaTeX.
- Instructor David integrates a single, course-long project into his computer science courses. By assigning a project title at the beginning of the term, "each lecture becomes a step closer to reaching the main project." This strategy provides a practical, unifying context for all theoretical lectures and lab sessions, helping students see the direct application of each new concept they learn.

These diverse examples demonstrate the flexibility of PBL and provide the practical basis for the activities outlined in the AIPLE framework, which synthesizes these successful strategies into a single, adaptable model.

*5.3. Navigating AI Integration: Between Cautious Optimism and Practical Hurdles*

Instructor perspectives on AI integration were nuanced, reflecting a blend of optimism about its potential and significant concerns about its practical and ethical implications. On the optimistic side, instructors see AI's potential for personalized learning, with adaptive systems that can analyze student performance and provide tailored recommendations. They also envision AI as a powerful tool for assisting students with research, generating diverse course materials, and even automating some assessment tasks. Instructor David, for example, personally uses AI tools like ChatGPT GPT-4o, ChatPDF2024, and web based Elicit for his own research to reformulate sentences and efficiently find relevant papers.

However, this optimism is tempered by significant practical and pedagogical hurdles. Instructor JNM provided a powerful cautionary tale from his direct experience using ChatGPT in a PBL project. The goal was for students to use AI to generate programming code and then modify and adapt it. Instead, "many students failed because they simply copied and pasted the code without making the necessary modifications." They did not take the time to understand the underlying logic, resulting in an exercise that produced "just engagement but not understanding or performance". This experience highlights the central risk identified by all instructors: AI can encourage cognitive offloading and undermine the development of critical thinking skills.

Beyond this pedagogical risk, instructors raised a host of other concerns, including:

- Infrastructure and Cost: High implementation costs and the fact that "not all students have access to high-performance computers and stable internet" create significant barriers to equitable access.

- Ethical Concerns: Instructors worry about accountability for incorrect AI-generated answers ("hallucinations"), the potential loss of human interaction between students and teachers, and the scientific rigor of unverifiable sources.
- Faculty Readiness: Instructors themselves require training to understand AI tools, their limitations, and how to adapt their teaching methods accordingly. As Instructor David noted, resistance is likely to come from educators who need to change long-held habits.

These deeply felt concerns underscore the need for a thoughtful, critical, and strategic approach to AI integration, a principle that lies at the heart of the AIPLE framework's design.

## 6. Discussion: Implications for Sustainable Engineering Education

### 6.1. Applying the AIPLE Framework to Overcome Foundational Learning Barriers

The AIPLE framework directly addresses the student apprehension and passive learning culture identified by the instructors. By grounding abstract theoretical concepts in tangible, real-world sustainability problems, the framework transforms students' relationship with foundational sciences from one of fear or boredom to one of empowered application. The problem-first approach provides an immediate "why" for learning, motivating students to seek out the theoretical knowledge they need to solve a practical challenge. This aligns with constructivist principles and directly counters the "recipe-seeking" mindset by presenting problems that have no single, pre-defined solution.

The framework's cyclical and collaborative design is particularly well-suited to the heterogeneous classroom. The hands-on experimentation and prototyping stage (Stage 4) allows students with strong technical backgrounds to excel and mentor their peers, while the collaborative investigation and concept mapping stages (Stages 2 and 3) provide a supportive environment for students with weaker theoretical foundations to build confidence and understanding through peer-to-peer learning. This iterative process, where students repeatedly encounter and apply concepts, provides multiple opportunities to master essential knowledge while maintaining engagement through practical, meaningful work.

### 6.2. A Call for Responsible AI Integration: Navigating Ethics and Infrastructure

The integration of AI into education in developing countries cannot be a simple matter of adopting tools developed in and for high-resource contexts. The significant infrastructural, financial, and pedagogical challenges demand a more critical and strategic approach. The AIPLE framework offers a model for such responsible integration, acting as a pedagogical "guardrail" that structures the use of AI to mitigate its risks while harnessing its benefits.

The primary pedagogical risk of generative AI, as identified by Instructor JNM's classroom experience, is its potential to facilitate superficial engagement and cognitive offloading, where students use the tool to achieve answers without developing understanding. The most crucial recommendation from the instructors was that any AI-generated output, whether it be code, calculations, or design concepts, must be verified through real-world testing and implementation. The AIPLE framework inherently enforces this principle. Stage 4, "Hands-on Experimentation and Prototyping," is a mandatory, non-negotiable part of the learning cycle. A student cannot successfully complete an AIPLE project by simply submitting an AI-generated report; they must use that output as a starting point for a design that is then simulated, built, tested, and validated. This embeds empirical validation and critical thinking directly into the workflow, forcing students to move beyond passive acceptance of AI outputs and engage with them as active, critical engineers.

This approach aligns with emerging global frameworks for AI education, which call for the development of AI literacy that goes beyond simple use to include evaluation, creation, and ethical application [30]. Furthermore, by emphasizing the use of open-source and

low-cost AI tools, the framework remains mindful of the resource constraints prevalent in the DRC and other developing nations, promoting a more equitable and sustainable model of technology integration. However, successful implementation still requires institutional commitment to address the larger issues of infrastructure, digital equity, and comprehensive professional development for faculty [31].

### 6.3. The Critical Role of Community and Stakeholder Engagement

For Engineering Education for Sustainable Development to be truly effective, the problems students solve must be authentic, relevant, and consequential. The AIPLE framework's emphasis on "real-world sustainability challenges" must therefore extend beyond textbook case studies to include genuine engagement with community stakeholders. Recent pedagogical literature calls for empowering engineering students to become socially conscious and responsible agents of change by connecting their learning to the needs of the communities they will one day serve [32].

This requires a re-imagining of Stage 1 of the AIPLE framework, "Problem Identification and Contextualization." Drawing on best practices for community engagement in engineering, this stage should be a collaborative, co-creative process rather than a top-down, instructor-led exercise. This involves identifying the right stakeholders (e.g., local community groups, neighborhood leaders, non-governmental organizations), developing a transparent engagement plan, and working with them to define problems that are meaningful to the community. Such an approach ensures that the engineering solutions developed are not only technically sound but also socially relevant and culturally appropriate. It transforms the learning experience from a purely academic exercise into a form of service-learning, where students apply their skills to make a tangible, positive impact, thereby fostering a deep and lasting commitment to sustainability.

## 7. Conclusions and Future Research

This study introduces the AI-Integrated Practical Learning in Engineering (AIPLE) Framework, a comprehensive pedagogical model designed to address the persistent challenges hindering sustainable engineering education in the sub-Saharan Africa. Grounded in qualitative data from experienced engineering instructors in the Democratic Republic of Congo, the research confirms that a passive learning culture, rooted in heterogeneous pre-university preparation, remains a significant barrier to developing the critical competencies required for sustainable development. The AIPLE framework directly confronts this challenge by synergizing Problem-Based Learning (PBL), hands-on experimentation, and strategic AI integration within a five-stage cyclical process. Its primary contribution is a context-adapted, theoretically robust model that is informed by the practical realities of resource-constrained environments. The findings demonstrate that while instructors have independently adopted diverse and effective PBL strategies, their perspectives on AI are marked by a cautious optimism, tempered by significant infrastructural, pedagogical, and ethical concerns. The AIPLE framework offers a structured pathway for responsible innovation by embedding AI within a pedagogical cycle that mandates empirical validation and critical thinking, thereby mitigating the risk of superficial technological engagement. By positioning authentic sustainability challenges at the core of the learning process, the framework provides a scalable and replicable solution for cultivating engineers who are not only technically proficient but also equipped with the systems-thinking and ethical reasoning necessary to advance the Sustainable Development Goals.

The validation and refinement of the AIPLE framework necessitate a multi-pronged agenda for future research. A critical next step is the empirical validation of the framework through quasi-experimental studies designed to quantitatively measure its impact on

student learning outcomes, including conceptual understanding, problem-solving capabilities, and the development of specific sustainability competencies. Such studies should be complemented by qualitative analyses of student perspectives to gain a holistic understanding of their learning experiences, engagement levels, and perceived career preparedness. Furthermore, comparative longitudinal studies are required to assess the framework's adaptability and effectiveness across diverse institutional and cultural contexts, tracking graduate career trajectories to provide definitive evidence of its long-term impact on professional practice and its contribution to sustainable development.

**Author Contributions:** Conceptualization, R.K.T. and A.S.-M.; methodology, R.K.T. and A.S.-M.; validation, D.K.M., T.N.K. and J.N.M.; formal analysis, A.S.-M.; writing—original draft preparation, A.S.-M.; writing—review and editing, R.K.T., D.K.M., T.N.K. and J.N.M.; visualization, R.K.T.; supervision, R.K.T.; project administration, R.K.T. All authors have read and agreed to the published version of the manuscript.

**Funding:** Anderson Sunda-Meya acknowledges support from the Xavier University/State of Louisiana Endowed Chair in Science.

**Institutional Review Board Statement:** The study was conducted in accordance with the Declaration of Helsinki, and the protocol was approved by the Research Ethics Committee (REC) of Université Loyola du Congo (ULC-REC-2023-0003) on 15 September 2023.

**Informed Consent Statement:** Informed consent was obtained from all subjects involved in the study.

**Data Availability Statement:** Data available upon request.

**Acknowledgments:** The authors acknowledge the participation of engineering instructors at Université Loyola du Congo who contributed their time and insights to this research. We also thank the students and community members who inspired the development of this framework through their engagement with sustainability challenges. Special recognition goes to the ULC administration for supporting innovative pedagogical research and the Xavier University of Louisiana for facilitating international collaboration in engineering education.

**Conflicts of Interest:** The authors declare no conflicts of interest.

## Abbreviations

The following abbreviations are used in this manuscript:

| | |
|---|---|
| AI | Artificial Intelligence |
| AIPLE | AI-Integrated Practical Learning in Engineering |
| DRC | Democratic Republic of Congo |
| EESD | Engineering Education for Sustainable Development |
| ICT | Information and Communication Technology |
| PBL | Problem-Based Learning |
| SDG | Sustainable Development Goal |
| ULC | Université Loyola du Congo |
| UN | United Nations |
| UNESCO | United Nations Educational, Scientific and Cultural Organization |
| USA | United States of America |

## Appendix A

*Survey Questions*

Survey Section 1: Teaching Practices
1. Traditional Methods:

- Could you walk us through your typical teaching process for an engineering course, including how you structure lectures, assignments, and lab sessions?

- How do you adapt your teaching for students who lack strong backgrounds in mathematics and physics?
- What are the main challenges you encounter when using traditional teaching methods in engineering education?

2. Current Innovations:

- Have you incorporated digital resources or online tools into your teaching? If so, what tools or platforms have you found most effective?
- How do you currently measure the balance between theoretical and practical learning in your courses?
- Are there specific teaching strategies you've tried that have noticeably improved student engagement or outcomes?

Section 2: Problem-Based Learning (PBL)

3. Implementation of PBL:

- When did you first start incorporating Problem-Based Learning (PBL) into your teaching? What motivated this shift?
- Could you describe a specific PBL activity or project you have used? What was the focus, and how was it structured?
- How do you ensure that students stay engaged and collaborative during PBL activities?

4. Impact on Students:

- What changes have you noticed in students' problem-solving skills, creativity, or teamwork abilities since implementing PBL?
- Have you observed any differences in how students approach engineering challenges when using PBL versus traditional methods?
- How do students typically respond to PBL activities, particularly those who might initially lack confidence in STEM subjects?

5. Challenges and Solutions:

- What logistical or resource-related barriers have you encountered when implementing PBL?
- How do you adapt PBL methods for large classes, limited lab access, or time constraints?
- Could you share an example of a challenge you faced with PBL and how you resolved it?

Section 3: Perspectives on AI Integration

6. Understanding and Usage of AI:

- What is your current understanding of Artificial Intelligence (AI), and how do you see its role in engineering education?
- Have you personally used any AI-based tools (e.g., virtual labs, adaptive learning platforms, simulations) in your teaching? If so, which ones, and what has been your experience with them?
- Do you actively teach students about AI concepts or applications in engineering? If yes, how is this incorporated into your curriculum?

7. Perceived Benefits:

- What specific areas of engineering education (e.g., labs, assessments, personalized learning) do you think AI has the most potential to improve?
- How do you think AI could help overcome barriers to education in resource-constrained settings?
- Have you observed any improvements in student engagement, understanding, or performance when using AI tools?

8. Concerns and Limitations:

- What challenges or limitations do you foresee in integrating AI into engineering education (e.g., cost, ethical concerns, technical barriers)?
- Are there any potential risks you see in over-relying on AI in the classroom?
- How would you address potential resistance from students or educators to adopting AI-based teaching tools?

Section 4: Service Learning (Optional)

9. Experience with Service Learning:

- Have you implemented service learning as part of your teaching? If so, what type of projects or partnerships have you facilitated?
- Could you provide an example of how service learning has helped students apply their engineering skills to real-world problems?
- How do you assess the impact of service learning on students' technical skills and their understanding of community needs?

Section 5: Future Directions

10. Suggestions for Improvement:

- What are the most significant gaps you see in current engineering education methods, particularly in resource-limited settings?
- What specific types of support (e.g., funding, training, infrastructure) would be necessary to implement an effective AI and PBL-integrated approach in under-resourced areas?
- Are there any policies or institutional changes you believe are critical to advancing engineering education in the Sub-Saharan Africa?

11. Vision for the Future:

- How do you envision engineering education evolving in the next decade, particularly in light of advancements in AI and other digital technologies?
- What steps can educators take to prepare students for addressing global challenges like climate change and sustainable development?
- How can engineering education better align with the UN's Sustainable Development Goals?

Follow-Up Probes for Depth

- "Can you provide a specific example?"
- "What tools or strategies were particularly effective in this scenario?"
- "How did students respond to this method or technology?"

This detailed structure ensures comprehensive coverage of the topics and will generate rich, actionable insights for your research. Let me know if you'd like to refine further!

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
