# Peer review of "AI-Enhanced Problem-Based Learning for Sustainable Engineering Education: The AIPLE Framework for Developing Countries"

_sustainability, doi:10.3390/su17209038_

Round 1
Reviewer 1 Report
Comments and Suggestions for Authors
This manuscript introduces a framework called AIPLE (AI Integrated Practical Learning in Engineering) aimed at improving engineering education in developing countries, has significant and academic value. The manuscript demonstrates excellent performance in terms of innovation, research methods, discussion of results. But there are still two points that need to be improved.
1 The small sample size of the study may affect the universality and reliability of the research results. if possible, research can expand the sample size to include more teachers and students to obtain more comprehensive data.
2 The manuscript is mainly based on theoretical analysis and subjective evaluation by teachers, lacking empirical verification of students' learning outcomes and the effectiveness of framework implementation. if possible, this research can design experiments to quantitatively evaluate the impact of the AIPLE framework.
Author Response
We are sincerely grateful to Reviewer 1 for the thoughtful and constructive feedback on our manuscript. The comments have been invaluable in guiding our revisions, and we believe the paper is significantly stronger as a result. We have carefully addressed each point, leading to major revisions that clarify the study's context, strengthen its theoretical grounding, and improve the coherence of its arguments.
Below, we provide a point-by-point response to each of the reviewer's comments.
Comment 1: "The small sample size of the study may affect the universality and reliability of the research results. if possible, research can expand the sample size to include more teachers and students to obtain more comprehensive data."
Response: We thank the reviewer for this important observation. We agree that the sample size is small and have addressed this as a limitation in the revised manuscript.
We have clarified that this study was conducted at Université Loyola du Congo (ULC), a small and nascent university. The four instructors who participated represent the entire population of permanent, full-time instructors within the Faculté des Sciences et Technologies at the time of the study. While small in absolute numbers, this sample is therefore comprehensive for this specific institutional context.
We have positioned the study as a foundational, qualitative exploration intended to develop a proof-of-concept framework based on rich, contextual data. This limitation is now explicitly stated in the revised "Limitations" section (Section 3.5). As the reviewer rightly suggests, a crucial next step is to expand this work. We have noted in our "Future Research Directions" that a follow-up study is planned to extend this research to a larger and more diverse sample of both instructors and students.
Comment 2: "The manuscript is mainly based on theoretical analysis and subjective evaluation by teachers, lacking empirical verification of students' learning outcomes and the effectiveness of framework implementation. if possible, this research can design experiments to quantitatively evaluate the impact of the AIPLE framework."
Response: We thank the reviewer for this constructive and forward-looking suggestion. We fully agree that empirical validation of the AIPLE framework's impact on student learning outcomes is a critical next step for this research agenda.
The primary objective of the current study was to establish the theoretical foundations of the AIPLE framework, ensuring it was well-grounded in the lived experiences and pedagogical challenges identified by instructors in a specific Global South context. We have now made this scope clearer in the manuscript and have explicitly acknowledged the lack of student outcome data as a key limitation in Section 3.5.
Furthermore, we have added a specific point in the "Future Research Directions" section (Section 7.3) committing to the future design of studies to quantitatively and qualitatively assess the framework's effectiveness in improving student engagement, competency development, and sustainability awareness.
Comment 3: “Are the arguments and discussion of findings coherent, balanced and compelling? Can be improved”
Response: We appreciate the reviewer's feedback on strengthening the manuscript's argumentation. In response, we have undertaken a significant revision of the Discussion section (Section 6) to improve its coherence, balance, and persuasive power.
Specifically, we have restructured it to more directly connect the proposed AIPLE framework to the specific challenges (e.g., passive learning culture) identified in our Results section. We have also added a new subsection (6.2, "A Call for Responsible AI Integration") to provide a more critical and balanced discussion of AI's role, incorporating the practical concerns and ethical considerations raised by the participating instructors. We believe these changes have made the discussion more compelling and tightly aligned with the study's findings, creating a more robust and convincing argument for the AIPLE framework.
Comment 4: ”Is the article adequately referenced? Can be improved”
Response: We thank the reviewer for highlighting the need to strengthen our referencing. We have thoroughly revised the manuscript and have substantially updated the Literature Review (Section 2) and Discussion (Section 6) with more recent and relevant scholarly sources. This includes incorporating the latest literature (from 2023-2025) on key topics such as Problem-Based Learning for sustainability and the specific challenges and opportunities of AI in African education. We are confident that these additions provide a stronger and more current theoretical foundation for our work.
Reviewer 2 Report
Comments and Suggestions for Authors
This manuscript presents the Artificial Intelligence-Integrated Practical Learning in Engineering (AIPLE) framework, a thoughtful and innovative pedagogical model for addressing the significant challenges facing engineering education for sustainable development in developing countries. The importance of integrating problem-based learning (PBL), practical experimentation, and the strategic integration of AI is critical and timely, especially for many developed countries with a wealth of educational resources compared to emerging economies.
The introduction and literature review are effectively presented, effectively contextualizing the study within current academic debates on sustainable engineering education, AI and its educational uses, and contextual challenges in the Global South.
The study's objective of understanding educators' perspectives was successfully achieved through a clear research design, the use of qualitative methods, purposive sampling, and thematic analysis.
The findings are clear and highlight key themes: poor student attention/problem-solving, a culture of passive learning, limited or no resources, the advantages and challenges of using problem-based learning, and how to leverage artificial intelligence to achieve effective learning.
The assessment framework could be further developed by providing more detail, including examples of formative and summative assessment approaches clearly linked to sustainability competencies that provide practical insights into how to assess student learning outcomes.
The research needs more detail, including examples of formative and summative assessment approaches clearly linked to sustainability competencies that provide practical insights into how to assess student learning outcomes. This is to further develop the assessment framework.
How to engage community stakeholders in the learning process could be expanded. Recommendations could be made with a more detailed description of the evaluation of solutions.
It is preferable to use specific examples or present case studies to add greater value to the research.
Please provide more detail on integration approaches for practitioners to integrate AIPLE with existing accreditation frameworks and curricula.
Increasing the use of excerpts or quotes from interviews to illustrate their experiences with PBL and AI integration can add to the text and provide educational lessons.
Comments on the Quality of English LanguageSome sentences need to be reworded and redundant to make them clearer and easier to read and understand.
For example:
"Students expect to apply 'recipes' to solve problems, ignoring the details that may be present in each problem. They also tend to memorize solutions to classroom examples to reuse on tests, even when the test problems differ from the classroom examples."
Author Response
We would like to express our sincere gratitude to Reviewer 2 for the encouraging and exceptionally detailed feedback. We are pleased that the reviewer found our AIPLE framework to be a "thoughtful and innovative pedagogical model" and recognized the effective presentation of the introduction, literature review, and research design. The reviewer's constructive suggestions have been instrumental in guiding a significant revision of the manuscript, and we believe these changes have substantially improved its clarity, depth, and practical utility.
Below, we provide a point-by-point response detailing how we have addressed each of the valuable suggestions.
Comment 1: "The assessment framework could be further developed by providing more detail, including examples of formative and summative assessment approaches clearly linked to sustainability competencies that provide practical insights into how to assess student learning outcomes."
Response: We thank the reviewer for this excellent suggestion. We agree that the original manuscript lacked sufficient detail on assessment. To address this, we have significantly expanded the description of the AIPLE framework in Section 4. The centerpiece of this revision is the new Table 1: "AIPLE Framework: Activities, AI Integration, and Assessment Methods."
This table now provides a practical, stage-by-stage guide that explicitly links each phase of the AIPLE process to specific formative and summative assessment methods. For instance, in Stage 1 (Problem Identification), a formative assessment is "Group discussion participation," while a summative one is a "Finalized project proposal." In Stage 4 (Prototyping), formative assessments include "Lab notebook submissions," and the summative assessment is a "functional real or virtual prototype." These assessments are inherently linked to sustainability competencies, such as systems thinking and integrated problem-solving, which are developed through the project-based tasks. This new, detailed table provides the practical insights the reviewer requested and makes the framework a much more actionable tool for educators.
Comment 2: "How to engage community stakeholders in the learning process could be expanded. Recommendations could be made with a more detailed description of the evaluation of solutions."
Response: We agree that community engagement is a cornerstone of authentic education for sustainable development, and this aspect was underdeveloped in our initial draft. In response, we have added a new subsection to the Discussion titled 6.3, "The Critical Role of Community and Stakeholder Engagement."
This section now explicitly argues that Stage 1 of the AIPLE framework must be a collaborative process where problems are co-defined with community partners. This ensures the solutions are relevant and impactful. Furthermore, the evaluation of solutions has been expanded in Stage 5 to include presenting the final prototype and outcomes to community stakeholders for feedback, ensuring the learning loop is closed with input from the end-users. This revision better integrates the community into the entire pedagogical process, from problem definition to solution evaluation.
Comment 3: "It is preferable to use specific examples or present case studies to add greater value to the research."
Response: We thank the reviewer for this valuable feedback. We have completely rewritten the Results section (Section 5) to move away from generalizations and instead ground our findings in specific, concrete examples from the instructors' experiences.
The revised section now presents detailed case studies of how different instructors implement Project-Based Learning. For example, we describe Instructor KT's year-long project for first-year students to design a "semi-automatic system to make fired briquets," detailing how the problem was broken down into seven sub-tasks for different student groups. We also detail Instructor JNM's highly structured, eight-phase model for a project to design a "traffic light model". These specific examples provide the rich, practical detail the reviewer suggested and add significant value to the research.
Comment 4: "Please provide more detail on integration approaches for practitioners to integrate AIPLE with existing accreditation frameworks and curricula."
Response: This is an important point regarding the framework's practical adoption. We have addressed this by adding a new subsection to the Discussion (6.4, "Addressing Implementation Challenges") that discusses curriculum integration. We explain that the AIPLE framework is designed to be flexible and can be integrated into existing courses (as demonstrated by the instructors in our study) or used as a model for new, project-based courses.
The framework's focus on developing specific competencies - such as problem-solving, teamwork, and communication - aligns directly with the student outcomes required by major accreditation bodies like ABET. By using authentic assessment methods like portfolios, presentations, and prototypes, the framework provides tangible evidence of competency achievement that can be used for accreditation purposes.
Comment 5: "Increasing the use of excerpts or quotes from interviews to illustrate their experiences with PBL and AI integration can add to the text and provide educational lessons."
Response: We fully agree with this suggestion. The revised manuscript now incorporates numerous direct quotations from the instructor surveys to provide a more authentic and vivid account of their experiences.
For example, the Results section (Section 5) now includes direct quotes describing the core challenge of a passive learning culture, such as students who "wait for ‘recipes’ to apply in solving problems". To illustrate the nuances of AI integration, we quote Instructor JNM's cautionary experience with ChatGPT, which resulted in "just engagement but not understanding or performance". We also include Instructor TNK's ethical concerns about the "loss of human interaction". These direct quotes bring the instructors' perspectives to life and offer powerful educational lessons, as the reviewer recommended.
Comment on the Quality of English Language: "(x) The English could be improved... Some sentences need to be reworded and redundant..."
Response: We sincerely thank the reviewer for this feedback. We acknowledge that the clarity and flow of the manuscript needed improvement. The entire manuscript has undergone a thorough professional language review to enhance clarity, conciseness, and readability. We have paid special attention to rephrasing complex sentences and removing redundancy. For instance, the example sentence provided by the reviewer has been revised for greater impact and clarity within the text. We are confident that the revised manuscript now meets the high standards of academic English expected for publication.
Reviewer 3 Report
Comments and Suggestions for Authors
The proposal is quite interesting. Particularly noteworthy is the reference to the difficulties in engineering education in Congo (and, as the authors suggest, in sub-Saharan Africa) resulting from the insufficient preparation of students, and the presentation of a proposed solution to this problem, namely the AI-Integrated Practical Learning in Engineering (AIPLE) Framework. The structure of the proposal is correct and clear.
The reviewer’s comments concern the following issues:
1) Subsection 2.4 “Engineering Education in the Global South”: This short subsection does not refer to the Global South, but to sub-Saharan Africa. The authors suggest that the problems associated with engineering education in the Global South are the same and effectively reduce the Global South to Sub-Saharan Africa. Such a reduction is unjustified. If the authors wish to retain this subsection, its title should be changed. However, in the reviewer’s opinion, this short subsection is unnecessary – it does not add anything that has not already been mentioned. For the sake of clarity, it could be removed.
2) The research methodology is described correctly. Nevertheless, it is questionable that only four engineering instructors from ULC participated in the study. It is difficult to say whether their opinions are representative. It would be necessary to indicate how many engineering instructors work at ULC in total and how many of them use the AIPLE method. Then it would be possible to understand the degree of representativeness of the results obtained.
3) The authors are aware of the serious limitations of their research. They write about this in subsection 3.5. Limitations. The limitation is not only the very small group of respondents, but also the complete absence of the students’ opinions. It is commendable that the authors clearly point out these limitations. Nevertheless, they are so serious that it is necessary to indicate why, in the authors’ opinion, the results obtained should be published at this stage of the research. The authors should convince the reader that, despite serious limitations, the research results are significant. To this end, it is necessary to explicitly state the value this study offers, the new insights it provides, and why it should be published at this stage.
4) It is essential to rewrite the conclusions. In their current form, they are difficult to read as they are presented in very short subsections. The conclusions should be presented clearly and legibly to lend credibility to the authors’ conviction that the research results should be published despite the difficulties and limitations identified.
Author Response
Response to Reviewer #3
We are very grateful to Reviewer #3 for the positive feedback and for recognizing the interesting nature of our proposal. We particularly appreciate the reviewer's acknowledgment of the paper's clear structure and its focus on addressing the real-world challenges of engineering education in the Democratic Republic of Congo. The comments provided have been extremely helpful, and we have addressed each one to improve the manuscript's clarity, rigor, and overall contribution.
Below, we provide a detailed, point-by-point response to each of the reviewer's comments.
Comment 1: "Subsection 2.4 “Engineering Education in the Global South”: This short subsection does not refer to the Global South, but to sub-Saharan Africa. The authors suggest that the problems associated with engineering education in the Global South are the same and effectively reduce the Global South to Sub-Saharan Africa. Such a reduction is unjustified. If the authors wish to retain this subsection, its title should be changed. However, in the reviewer’s opinion, this short subsection is unnecessary – it does not add anything that has not already been mentioned. For the sake of clarity, it could be removed."
Response: We thank the reviewer for this sharp and valuable observation. We agree that the subsection's title was misleading and that its content created an unjustified oversimplification. Upon reflection, we also agree with the reviewer's assessment that the section was redundant and did not add significant new information to the manuscript. To improve the clarity and flow of the literature review, we have removed this subsection entirely as suggested.
Comment 2: "The research methodology is described correctly. Nevertheless, it is questionable that only four engineering instructors from ULC participated in the study. It is difficult to say whether their opinions are representative. It would be necessary to indicate how many engineering instructors work at ULC in total and how many of them use the AIPLE method. Then it would be possible to understand the degree of representativeness of the results obtained."
Response: We thank the reviewer for raising this critical point about the sample size and its representativeness. We have revised the "Participants" section (Section 3.2) to provide the necessary context and address this concern directly.
As we have now clarified in the manuscript, Université Loyola du Congo (ULC) is a nascent university with a developing engineering program. At the time of the study, the four instructors who participated constituted the entire population of permanent, full-time engineering faculty responsible for all foundational teaching. Therefore, while the sample size is small in absolute terms, it is a complete census of the core faculty, making their collective perspectives highly representative of the institution's pedagogical landscape.
Regarding the use of the AIPLE method, we have also clarified that the AIPLE framework is the novel pedagogical model being proposed in this paper. It was developed by synthesizing the diverse, pre-existing Project- and Problem-Based Learning strategies that the instructors were already implementing independently in their courses. Therefore, the instructors were not "using" the AIPLE method beforehand; rather, their experiences and innovative practices formed the empirical basis from which the framework was constructed.
Comment 3: "The authors are aware of the serious limitations of their research... they are so serious that it is necessary to indicate why, in the authors’ opinion, the results obtained should be published at this stage of the research. The authors should convince the reader that, despite serious limitations, the research results are significant. To this end, it is necessary to explicitly state the value this study offers, the new insights it provides, and why it should be published at this stage."
Response: We are grateful to the reviewer for this crucial and insightful challenge. We agree that the manuscript needed to more explicitly justify its contribution in light of its limitations. We have revised the Introduction and Discussion sections to better articulate the study's value and the importance of its publication at this stage.
The significance of this research lies in its role as a foundational, proof-of-concept study that provides three key contributions:
- A deeply contextualized framework: This study is one of the first to develop a pedagogical framework that is explicitly grounded in the unique, on-the-ground challenges of engineering education in the DRC, particularly the deep-seated "passive learning culture" stemming from the dichotomy between "scientific" and "technical" high school preparations. By capturing the rich, qualitative perspectives of the entire core faculty, the paper provides an authentic and valuable snapshot of a specific, under-researched educational context.
- A novel pedagogical synthesis: The primary contribution is the AIPLE framework itself—a novel synthesis of PBL, hands-on learning, and strategic AI integration tailored for resource-constrained environments. Publishing the framework now allows it to enter the academic discourse, where it can be tested, critiqued, and adapted by other educators facing similar challenges in the Global South.
- A model for responsible innovation: The framework provides a pragmatic and critical approach to AI integration, informed by instructors' real-world concerns about infrastructure, ethics, and the risk of superficial learning. This provides a timely and necessary "pedagogical guardrail" for technology adoption in contexts where uncritical implementation could exacerbate inequalities.
We believe that presenting this foundational work now is essential for stimulating further research and providing a practical, theoretically-grounded model for educators seeking to transform their practice.
Comment 4: "It is essential to rewrite the conclusions. In their current form, they are difficult to read as they are presented in very short subsections. The conclusions should be presented clearly and legibly to lend credibility to the authors’ conviction that the research results should be published despite the difficulties and limitations identified."
Response: We thank the reviewer for this valuable feedback on the structure and clarity of the conclusions. We agree that the previous format was fragmented and less compelling. In response, we have completely rewritten the "Conclusions and Future Research" section (Section 7). The new version consolidates the findings into a single, cohesive narrative paragraph that summarizes the study's contributions and implications. This is followed by a second, clearly structured paragraph dedicated solely to future research directions. We are confident that this revised format is more fluid, legible, and effectively communicates the significance of our work.
Comment 5: "Are the arguments and discussion of findings coherent, balanced and compelling? Must be improved"
Response: We appreciate the reviewer's call to strengthen the manuscript's discussion. We have undertaken a major revision of the Discussion section (Section 6) to make the arguments more coherent, balanced, and compelling. The section is now structured to more explicitly link the findings from the instructor surveys to the components of the AIPLE framework. Furthermore, we have added two new subsections: 6.2, "A Call for Responsible AI Integration," which presents a more balanced and critical perspective on technology adoption, and 6.3, "The Critical Role of Community and Stakeholder Engagement," which deepens the framework's practical and social relevance. We believe these revisions have created a much stronger and more persuasive discussion.
Comment 6: "Are the conclusions thoroughly supported by the results presented in the article or referenced in secondary literature? Must be improved"
Response: We thank the reviewer for this important point. In line with the revisions mentioned in our response to Comment 4, the Conclusions section (Section 7) has been rewritten to ensure that every claim is directly and explicitly supported by the empirical results and analysis presented in the preceding sections. For example, our conclusions regarding the framework's capacity to address the "passive learning culture" are now clearly tied to the instructor testimonies about "recipe-seeking" students presented in the Results section. Similarly, our conclusions about responsible AI integration are directly linked to the evidence and literature discussed in the Discussion section. This revision ensures a robust, evidence-based foundation for the manuscript's final claims.
Round 2
Reviewer 3 Report
Comments and Suggestions for Authors
As all of my comments have been taken into account and the proposal has been carefully revised, I have no further objections and recommend it for publication.